# Copper-Iron Bimetal Ion-Exchanged SAPO-34 for NH$_3$-SCR of NO$_x$

**Tuan Doan** [1] , **Phong Dam** [1], **Khang Nguyen** [1] , **Thanh Huyen Vuong** [1,2,]*, **Minh Thang Le** [1] and **Thanh Huyen Pham** [1,]*

1 School of Chemical Engineering, Hanoi University of Science and Technology, 1 Dai Co Viet, Hanoi 100000, Vietnam; tuan.lochoadaub.k54@gmail.com (T.D.); phongdlq@gmail.com (P.D.); nnkhang711@gmail.com (K.N.); thang.leminh@hust.edu.vn (M.T.L.)

2 Leibniz Institute for Catalysis, University of Rostock, Albert-Einstein-str. 29a, 18059 Rostock, Germany

* Correspondence: huyen.vuong@catalysis.de (T.H.V.); huyen.phamthanh@hust.edu.vn (T.H.P.); +49-381-1281-410 (T.H.V.); Tel.: +84-986-986-988 (T.H.P.)

**Abstract:** SAPO-34 was prepared with a mixture of three templates containing triethylamine, tetraethylammonium hydroxide, and morpholine, which leads to unique properties for support and production cost reduction. Meanwhile, Cu/SAPO-34, Fe/SAPO-34, and Cu-Fe/SAPO-34 were prepared through the ion-exchanged method in aqueous solution and used for selective catalytic reduction (SCR) of NO$_x$ with NH$_3$. The physical structure and original crystal of SAPO-34 are maintained in the catalysts. Cu-Fe/SAPO-34 catalysts exhibit high NO$_x$ conversion in a broad temperature window, even in the presence of H$_2$O. The physicochemical properties of synthesized samples were further characterized by various methods, including XRD, FE-SEM, EDS, N$_2$ adsorption-desorption isotherms, UV-Vis-DRS spectroscopy, NH$_3$-TPD, H$_2$-TPR, and EPR. The best catalyst, 3Cu-1Fe/SAPO-34 exhibited high NO$_x$ conversion (> 90%) in a wide temperature window of 250–600 °C, even in the presence of H$_2$O. In comparison with mono-metallic samples, the 3Cu-1Fe/SAPO-34 catalyst had more isolated Cu$^{2+}$ ions and additional oligomeric Fe$^{3+}$ active sites, which mainly contributed to the higher capacity of NH$_3$ and NO$_x$ adsorption by the enhancement of the number of acid sites as well as its greater reducibility. Therefore, this synergistic effect between iron and copper in the 3Cu-1Fe/SAPO-34 catalyst prompted higher catalytic performance in more extensive temperature as well as hydrothermal stability after iron incorporation.

**Keywords:** Cu-Fe/SAPO-34; NH$_3$-SCR; NO reduction; bimetal liquid ion-exchanged

## 1. Introduction

Despite the advantages of good fuel economy and high-power density, modern diesel engines working under lean combustion conditions release several types of air pollutants, including carbon oxides (CO$_x$), sulfur oxides (SO$_x$), volatile organics (VOCs), and especially nitrogen oxides. NO$_x$ is an abbreviated word for various kinds of nitrogen oxides, which primarily relates to nitrogen dioxide (NO$_2$) and nitrogen monoxide (NO) [1,2]. NO$_x$ can have serious human health and environmental effects, such as acid rain, photochemical smog, global warming, nose and eye irritation, respiratory diseases, etc. [3]. Those negative effects have drawn remarkable attention for researchers to figure out a solution to control NO$_x$ concentration, particularly from diesel engines exhaust. When it comes to NO$_x$ reduction technologies, selective catalytic reduction of NO$_x$ by ammonia (NH$_3$-SCR) is commonly considered to have the highest efficiency for NO$_x$ treatment from diesel engines. After the treatment, the exhaust fume can meet current tightening emission standards, such as European Emission Standards VI (EURO VI), and Super Ultra-low Emissions Vehicle (SULEV) [4–6]. In this technology, the catalyst

plays an important role and has many specific requirements. For example, the catalyst should either work in a wide operational temperature range (i.e., from 100 to 600 °C) to be suitable for both light and heavy diesel engine exhaust or reduce $NO_x$ selectively to $N_2$. In addition, it would also need to withstand the water vapor and sulfur oxides from any hydrocarbon combustion effluents [7]. Therefore, many efforts have been made to enhance the efficiency of catalysts for the SCR method.

Recently, some transition metals (Cu, Fe, Mn, and Ce) supported on zeolites have been used as catalysts for removing $NO_x$ at some typical range of temperature. The catalytic efficiency for $NH_3$-SCR of Cu/ZSM-5 and Cu/Beta has been proved at the standard temperature of engine exhaust (above 250 °C) [8,9]. However, the appearance of moisture during the reaction at a high temperature negatively affected to the catalysis by dealumination in the structures of Cu/ZSM-5 and Cu/Beta. Therefore, intensive studies on catalysts with high hydrothermal stability have been conducted and discovered a new family of SCR catalysts—Cu-chabazite (Cu-CHA) zeolites such as Cu/SSZ-13 and SAPO-34 [10–13]. On the other hand, Mn-based catalysts show a high activity at low temperatures (i.e., from 150 to 250 °C), which is the typical temperature of light-duty diesel engines and even advanced diesel engines exhaust [14,15]. However, the destruction of catalysts with the presence of moisture at the temperature of the soot and ash filter regeneration of a diesel particulate filters (DPF) (often over 450 °C) limits the catalytic ability of the currently commercialized catalysts for $NH_3$-SCR on diesel engines under this condition [16,17]. Therefore, the requirement for being active at high temperature could be crucial for SCR catalysts. To qualify for that requirement, a combination of multiple metals that have good performances at different ranges of temperatures could be an alternative solution. Meanwhile, Fe-zeolites have shown greater performance for $NH_3$-SCR at high temperatures of above 450 °C [18,19]. The layered catalyst concept was first introduced by Metkar et al. [20] and exhibited a high $NO_x$ conversion in a wide working temperature window. In this concept, a thin Fe-zeolite layer was used to cover a thick Cu-zeolite layer below. Another concept was developed with a long Cu-chabazite brick sequentially arranged with a thin layer of Fe/ZSM-5. Two-component systems such as Cu-CHA/Fe-SAPO-34 and Cu-SSZ-13/Fe-SSZ-13 were also reported by Gao et al. [9] and Andonova et al. [16] to achieve high $NO_x$ removal efficiency over a broad temperature range. Besides, Fe co-doped with Cu-CHA zeolite catalyst was also developed as a more direct method for synthesizing deNOx catalysts. The study of Zhang et al. [21] also shows that $Fe_xCu$/SSZ-13 catalyst synthesized with an ion-exchange method has a better deNOx efficiency at a high temperature than Cu/SSZ-13. Yang et al. [22] suggested that the catalytic performance of Cu/SSZ-13 can enhance at both low- and high-temperature conditions due to the presence of iron metal.

Metals loading on the catalysts mainly attributes to the catalytic performance while the carrier takes main responsible for the stability and selectivity production. Due to their already widespread use in related catalysis, easy availability, and frame stability at different operating temperatures, zeolites have recently attracted attention for many researchers. It was seen that the smaller pore the zeolite has, the more active the catalyst is. Among those small pore zeolite, SAPO-34 with a chabazite topology was reported to show greater hydrothermal stability, then SSZ-13 or ZSM-5 [23]. The hydrothermal stability was needed when the temperature of the exhaust reaches above 650 °C with the moisturized environment during the regeneration of DPF section, which is generally placed in front of the SCR section. SAPO-34 molecular sieves with low silicon content and uniform distribution are important for maximizing the selectivity of nitrogen [24].

Regarding the acidic properties of the catalysts, the mechanisms of formation of Brønsted acid sites of SSZ-13 and SAPO-34 are different in each catalyst. While Brønsted acid sites of SSZ-13 solely depend on Al content, these acid sites of SAPO-34 are mainly attributed to the substitution of Si atoms for P atoms in the neutral $AlPO_4$-34 framework [25]. H/SSZ-13 was proved both experimentally and theoretically to possess stronger acidity than the H/SAPO-34, resulting in various states of copper in the metal exchanged catalysts [26]. In the case of SAPO-34, a small number of copper oxides $(CuO_x)$ species coexisting with isolated $Cu^{2+}$ ions can lead to a reduction in the activity at high temperatures [27,28]. Copper species, on the other side, mainly occur in Cu/SSZ-13 structures as

isolated $Cu^{2+}$ ions [20]. Moreover, SAPO-34 can provide many different strong acid sites that are important for SCR activity [12]. Besides, our previous investigation also demonstrated that SAPO-34, using a mixture of Morpholine/Triethylamine (TEA)/Tetraethylammonium hydroxide (TEAOH) as organic structure-directing agents (OSDAs), showed high surface area, decreased the crystal size of the samples to nano-size, improved the crystallinity, and enhanced the acid properties [29].

Herein, we aimed to investigate the application of copper-iron bimetal ion-exchanged SAPO-34 (Cu-Fe/SAPO-34) for deNOx with $NH_3$ reductant and compare to single metal (Cu/SAPO-34 and Fe/SAPO-34). The SAPO-34 samples were prepared by using combinations of templates (TEA, Morpholine, and TEAOH), followed by last our report [29]. All the metal catalysts were prepared by aqueous ion-exchange method. The physicochemical properties of the Cu-Fe/SAPO-34, Cu/SAPO-34, and Fe/SAPO-34 were characterized by several methods. The primary mission concentrates on the following contents:

- The catalyst's properties of synthesized samples to clarify the influence of Cu, Fe, and co-doping Cu-Fe supported on SAPO-34.
- Additionally, the catalytic performance of the $NH_3$-SCR reaction, as well as the hydrothermal durability of the catalysts, was tested in this study.

## 2. Results and Discussion

### 2.1. Structure and Texture of Catalysts

The structure of catalysts investigated by the X-ray powder diffraction (XRD) method presented in Figure 1 shows that the characteristic properties of the prepared Cu/SAPO-34, Fe/SAPO-34, and Cu-Fe/SAPO-34 samples nearly coincide with as-prepared SAPO-34 sample. Specifically, the characteristic peaks for the CHA framework structure at $2\theta$ = 7.5, 11.1, 14.9, 19.8, 21.1, 22.5, 25.9, and 30.1° appeared in these samples. This confirms that all catalysts are similar to the typical chabazite structure of SAPO-34 reported previously [30], indicating that the typical structure of SAPO-34 was maintained during the aqueous ion-exchange process. Therefore, the crystallize sizes of metal-promoted zeolites are close in the range of 34–39 nm (Table 1). In addition, the peaks intensity decreases after doping metals, particularly with increasing the amount of Cu and Fe. Thus, this influence could consolidate the hypothesis of copper and iron species at the external surface of SAPO-34 [31] or decreased crystallinity (Table 1). For Cu–Fe/SAPO-34 catalysts, their peak intensity and corresponding crystallinity slightly higher than that of Cu/SAPO-34. This finding suggests that the simultaneous introduction of Cu and Fe onto SAPO-34 can reduce the effects of single metal on SAPO-34. Besides, metal oxides species such as CuO, $Cu_2O$, or $Fe_xO_y$ were not observed even at a higher metal content of 5 wt.%, indicating that metal species mainly exist in the form of isolated ions or well-distributed crystal metal oxides particles in the SAPO-34 structure [31,32].

**Table 1.** Physicochemical properties and crystallinity of the as-synthesized samples.

| Catalyst | Metal Atomic Concentration (wt.%) [a] | BET Surface (m²/g) [b] | Average Pore Diameter (nm) [b] | Mean Crystallite Size (nm) [c] | Crystallinity (%) |
|---|---|---|---|---|---|
| SAPO-34 | - | 697 | 0.75 | 34.29 | 77.48 |
| 1Cu/SAPO-34 | 1.07 | 583 | 0.69 | 34.77 | 75.06 |
| 3Cu/SAPO-34 | 3.12 | 501 | 0.60 | 35.64 | 72.29 |
| 5Cu/SAPO-34 | 4.89 | 357 | 0.47 | 36.76 | 67.68 |
| 1Fe/SAPO-34 | 0.99 | 555 | 0.65 | 34.60 | 74.04 |
| 3Fe/SAPO-34 | 2.91 | 487 | 0.60 | 38.30 | 71.94 |
| 5Fe/SAPO-34 | 4.78 | 332 | 0.44 | 39.43 | 63.67 |
| 2Cu-2Fe/SAPO-34 | 2.02 Cu–1.82 Fe | 479 | 0.53 | 38.31 | 72.82 |
| 3Cu-1Fe/SAPO-34 | 3.03 Cu–0.91 Fe | 499 | 0.56 | 34.71 | 73.87 |

[a] determined by EDS, [b] calculated from $N_2$ adsorption-desorption data and [c] derived by the Scherrer equation from XRD data.

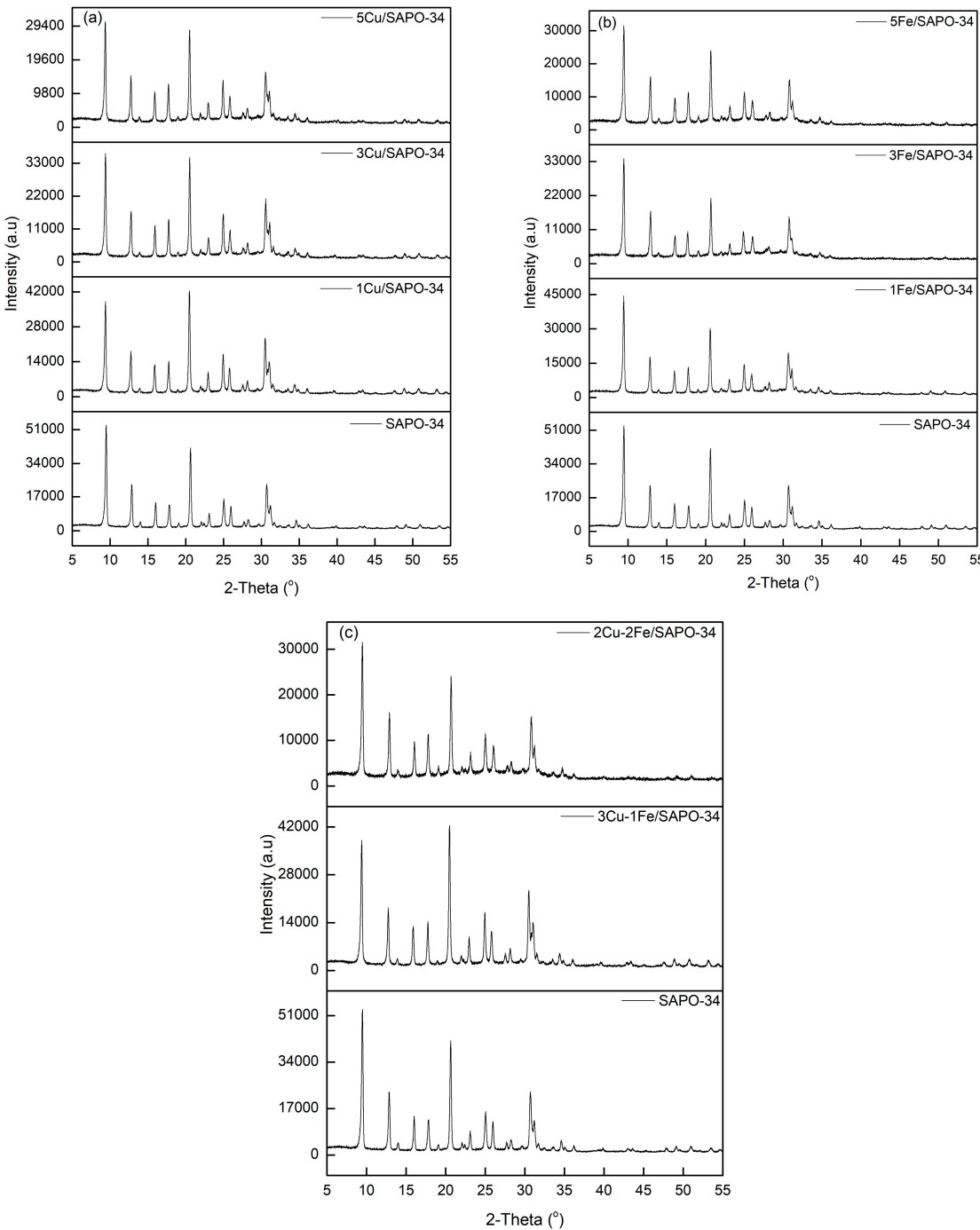

**Figure 1.** X-ray powder diffraction patterns of as-synthesized SAPO-34 and catalyst (**a**) Cu/SAPO-34, (**b**) Fe/SAPO-34, and (**c**) Cu-Fe/SAPO-34 with various metal content.

Table 1 shows the results of the textural properties of the synthesized samples investigated by the nitrogen adsorption isotherms method. The final metal loading in all samples determined by energy dispersive X-ray spectroscopy (EDS) analysis is close to the theory metal content calculated for catalyst preparation. It can be observed that both Brunauer-Emmett-Teller (BET) surface area and pore size slightly reduce when the metal loading varies in the range of below 3 wt.%, but the physical structure of SAPO-34 is significantly influenced by the doping of Cu and Fe at 5 wt.%. In general, a gradual decrease in BET surface area and micropore diameter depend on the content and type of metals. In order to avoid the reduction of BET surface area and achieve a high metal loading comparable to

mono-metallic zeolites, the total metal loading of bimetallic samples of 4 wt.% was chosen. As seen in Table 1, the BET surface area and pore diameter of both Cu–Fe/SAPO-34 catalysts (containing about 4 wt.% Cu and Fe) is similar to those of the two other single metal-containing catalysts with less metal content of about 3 wt.%.

　　Furthermore, the XRD results indicate that the co-doping of Cu and Fe reduces the effects of the metal introduction into the support SAPO-34 on its structure and properties. The $N_2$ adsorption-desorption plots in Figure 2 show type I isotherms for SAPO-34 and series of as-synthesized catalysts according to the International Union of Pure and Applied Chemistry (IUPAC) classification, indicating that all samples contained microporous structure [29]. It can also be noted the presence of a small hysteresis. As such, micropores are kept in all as-synthesized catalysts.

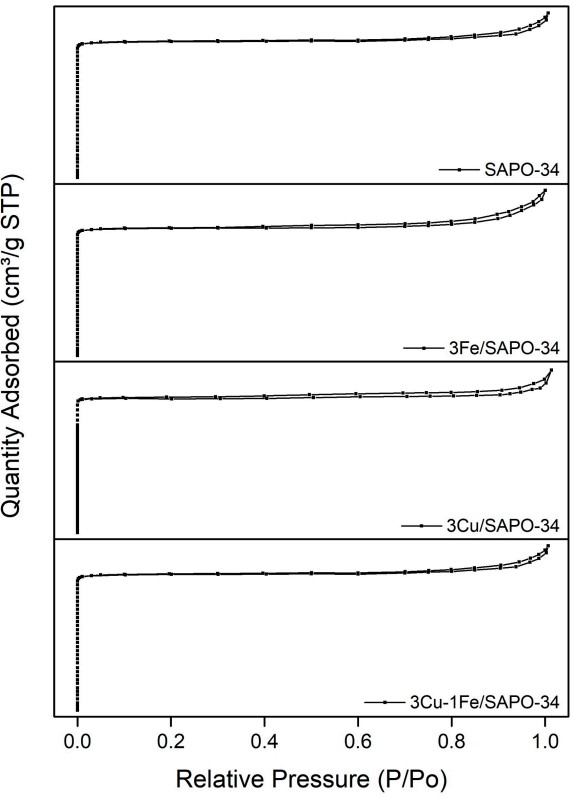

**Figure 2.** Nitrogen adsorption-desorption isotherms of as-synthesized samples.

　　Figure 3 illustrates the field emission scanning electron microscopy (FE-SEM) images of synthesized samples prepared by pure SAPO-34 and modified with different metals. As-synthesized samples display standard cubic SAPO-34 polycrystals of approximately 1–3 μm dimensions. The morphological features of the Cu/SAPO-34, Fe/SAPO-34, and Cu-Fe/SAPO-34 catalysts with different types of metals were compared with those SAPO-34. It can be found that all the aggregates of all samples roughly maintain the same cubic crystals of the metal-free materials, which suggest that the addition of Cu and Fe did not change the CHA structure and crystal morphology of SAPO-34. However, many scraps appeared on the external surface of the catalyst samples, especially in the high metal loading amount. As reported by Nana Zhang et al., some layers of suspended matters can be ascribed to the aggregated $Cu_xO$ nanoparticles [33]. That statement is also observed in Fe-exchanged samples that these random-shaped pieces assign to $Fe_xO_y$ clusters. Combined with EDS and XRD results, this can be attributed that some large metal-containing agglomerations are well-distributed crystal metal oxides particles in the SAPO-34 structure. In this case, Cu–Fe bimetal catalysts based on SAPO-34 containing about 4% metal, FE-SEM images showed a cubic-like rhombohedral morphology, and the catalysts are smoother than the two other single metal-promoted zeolites with more than 3% metal loading.

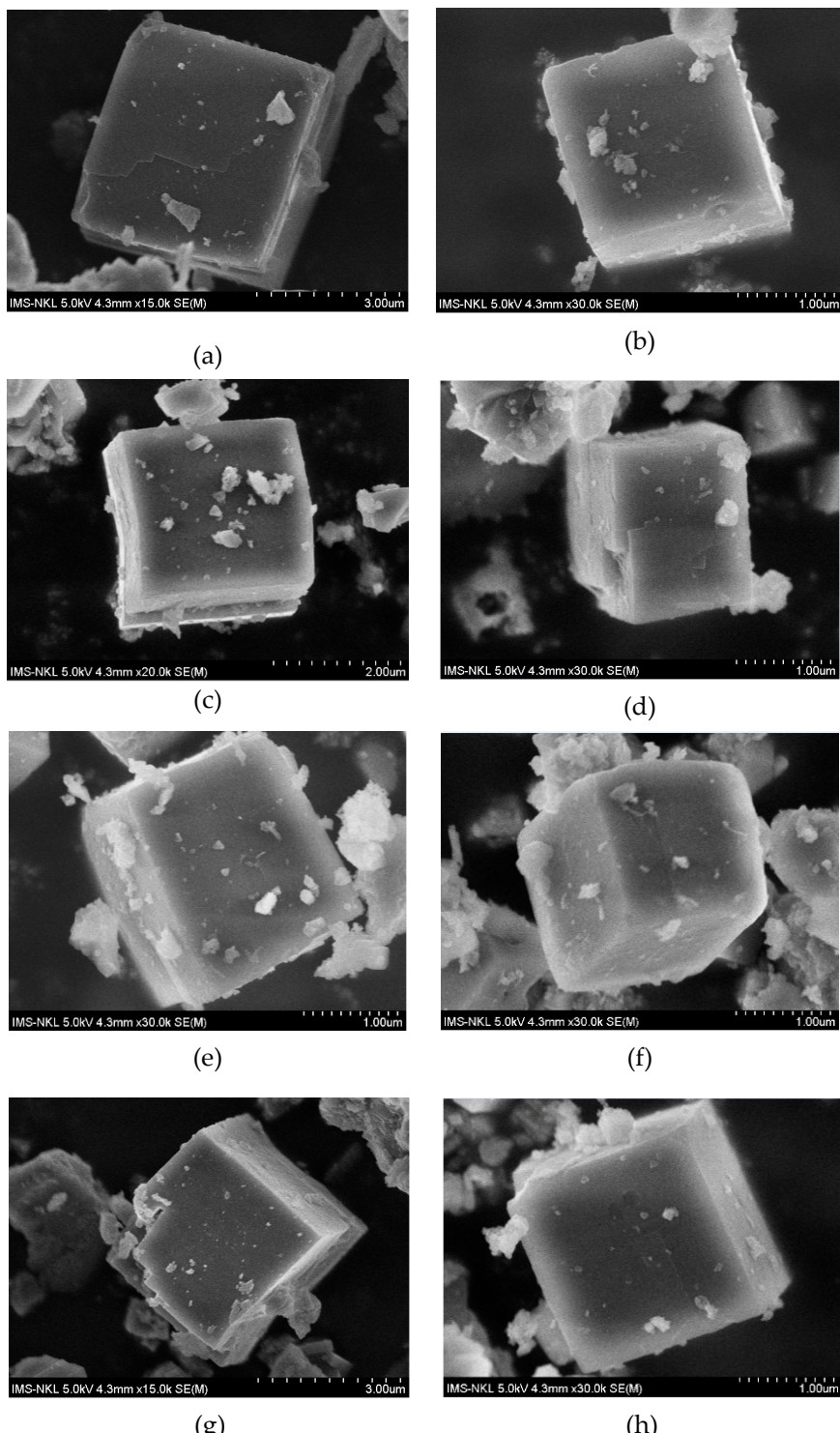

**Figure 3.** FE-SEM images of (**a**) 1Cu/SAPO-34, (**b**) 1Fe/SAPO-34, (**c**) 3Cu/SAPO-34, (**d**) 3Fe/SAPO-34, (**e**) 5Cu/SAPO-34, (**f**) 5Fe/SAPO-34, (**g**) 2Cu-2Fe/SAPO-34, (**h**) 3Cu-1Fe/SAPO-34.

## 2.2. Chemisorption Results

The distribution of acidic centers is illustrated by the temperature-programmed desorption with ammonia (NH$_3$-TPD) profile in Figure 4 and Table 2. Ammonia molecules desorbed at higher temperatures characterize stronger interactions, and hence stronger acidic sites. SAPO-34 samples exhibit both weak and strong acidic sites whose desorption peaks, respectively, appear at 150–200 °C

and 350–420 °C [34]. For the Cu/SAPO-34 catalyst, three desorption peaks are observed at around 150–200 °C, 270–400 °C, and 480–530 °C. The first one is ascribed to adsorbed at the weak acid sites by physiosorbed NH$_3$ or weak ammonium species. The next two peaks can be assigned to NH$_3$ adsorbed at the medium and strong acid sites, respectively [35]. The too-broad desorption peaks at around 400 °C indicate that the amount of Brønsted acid centers was still relatively high in the sample Cu/SAPO-34 [36,37]. Meanwhile, for the catalyst Fe/SAPO-34, one of the desorption peaks around 150–200 °C is assigned to physically adsorbed ammonia, and the higher temperature peaks at 350–420 °C, and 530–580 °C are due to ammonia desorbed from moderate and strong acid sites, respectively. However, compared to Cu/SAPO-34, the desorption peak position at 530–580 °C for strong acid sites of Fe/SAPO-34 shifted to a higher temperature by about 50 °C, indicating that the strength of these sites in Cu/SAPO-34 was lower than those in the Fe/SAPO-34. Increasing Cu or Fe content to 5 wt.% improves their capacity of NH$_3$ adsorption since their peak area reaches the largest values (Table 2), and the peak temperatures, especially medium acid sites, are a wider temperature range compared to that of lower loading samples. The catalyst co-doping with a mixture of Cu-Fe caused the formation of three desorption peaks, which are broader and higher intensity than those of Cu/SAPO-34, especially the weak and moderate acid sites. This indicates the considerable number of acid sites of Cu–Fe/SAPO-34 catalysts [38]. In our situation, the combination of the Fe(CH$_3$COO)$_2$·4H$_2$O with the Brønsted acid protons in the Cu/SAPO-34 make the acidic aqueous environment during preparation process, resulting in the formation of extra-framework Cu$^{2+}$ from a part of bulk CuO [39]. Moreover, a previous study [31] suggested that Cu exchanged to SAPO-34 is able to improve Lewis acidity by substitute H$^+$ with Cu$^{x+}$. An increase in the exchange capacity of Cu, therefore, can enhance the Lewis acid strength of Cu–Fe/SAPO-34, leading to a higher amount of adsorbed NH$_3$ molecules on the catalyst.

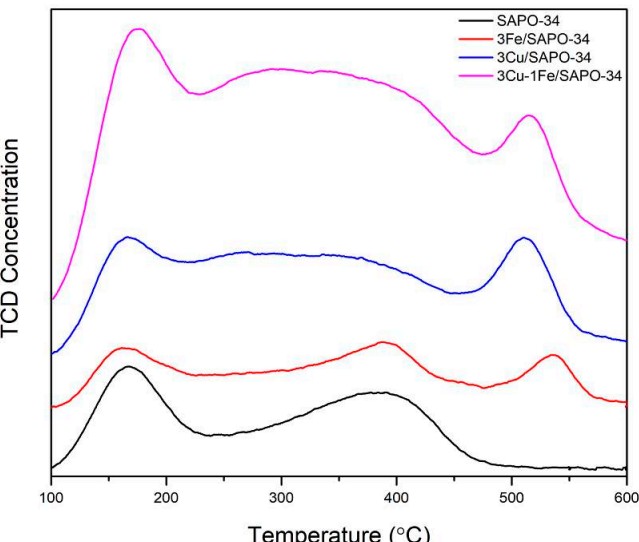

**Figure 4.** Temperature-programmed desorption of ammonia as-synthesized samples.

**Table 2.** Acid properties of the as-synthesized catalyst.

| Samples | Peak (mmol/g) | | | Total (mmol/g) |
|---|---|---|---|---|
| | Peak 1 | Peak 2 | Peak 3 | |
| SAPO-34 | 1.22 | 0.98 | - | 2.2 |
| 1Cu/SAPO-34 | 1.35 | 2.97 | 1.32 | 5.64 |
| 3Cu/SAPO-34 | 1.82 | 4.45 | 2.41 | 8.68 |
| 5Cu/SAPO-34 | 3.22 | 6.02 | 2.93 | 12.17 |
| 1Fe/SAPO-34 | 1.05 | 2.11 | 1.36 | 4.52 |
| 3Fe/SAPO-34 | 1.69 | 2.81 | 1.79 | 6.29 |
| 5Fe/SAPO-34 | 1.97 | 3.18 | 1.96 | 7.11 |
| 2Cu-2Fe/SAPO-34 | 1.91 | 3.02 | 1.89 | 6.82 |
| 3Cu-1Fe/SAPO-34 | 2.92 | 5.34 | 2.47 | 10.73 |

Furthermore, in order to determine the redox properties of the catalysts, three samples (3Cu/SAPO-34, 3Fe/SAPO-34, and 3Cu-1Fe/SAPO-34) were selected and measured by temperature-programmed reduction with $H_2$ ($H_2$-TPR) measurements, whose results are presented in Figure 5. As shown in Figure 5, all catalysts exhibit two broad peaks at low temperatures (below 500 °C) and at high temperatures (above 550 °C). The $H_2$ consumption peak at around 365 °C can be assigned to the reduction of $Fe^{3+}$ to $Fe^{2+}$ in isolated ions and clusters, while the peaks at 440 °C and 660 °C are related to oligomeric iron species and the reduction of $Fe^{2+}$ to $Fe^0$ in oligomeric clusters [40,41]. The $H_2$-profile of Cu/SAPO-34 catalyst show two reduction peaks at different temperature ranges of 150–400 °C and 600–800 °C, which describe the two-stage reduction of isolated $Cu^{2+}$ ($Cu^{2+}$ to $Cu^+$ and $Cu^+$ to $Cu^0$). The first peak can be deconvoluted into three $H_2$ consumption peaks with tops at 215, 255, and 325 °C, respectively, which are related to the reduction of different $Cu^{2+}$ sites. In details, the peak at 215 °C can be assigned to the reduction of isolated $Cu^{2+}$ ions, while the peak at 255 °C and 325 °C are attributed to the reduction of unstable $Cu^{2+}$ inside the cage of chabazite structure, and the reduction of stable $Cu^{2+}$ located in six-membered rings, respectively [42]. The peak at above 550 °C is formed by consuming $H_2$ during the reduction of $Cu^+$ to $Cu^0$ at the cation exchange centers [43,44].

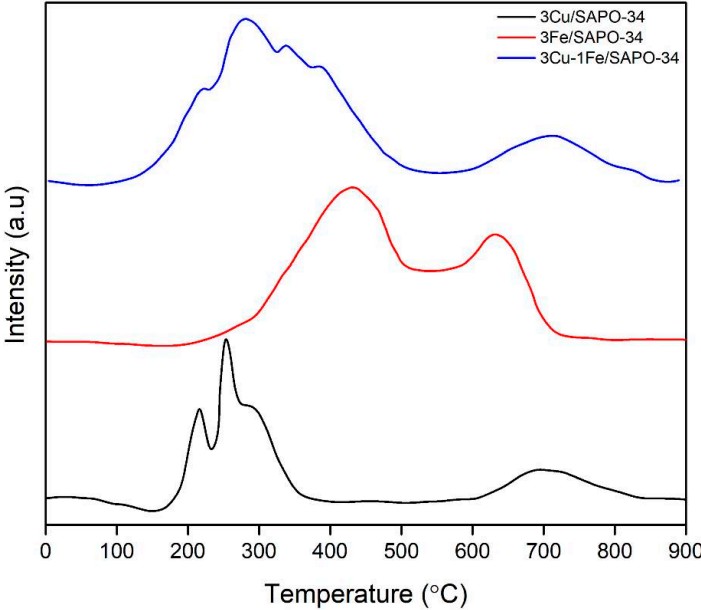

**Figure 5.** $H_2$-TPR profiles of catalyst samples.

The co-doping of Cu and Fe into SAPO-34 leads to shifting to the higher temperature of the first reduction peak compared to Cu/SAPO-34. This is due to the intense interaction between copper and

iron metallic components and integration in Cu-Fe/SAPO-34, making them more difficult to reduce. The first peak at 225 °C is attributed to the reduction of surface bulk CuO species and isolated $Cu^{2+}$ ions [45,46] while the peaks centered at 280 °C and 335 °C represent the reduction of $Cu^{2+}$ sites located in different positions of the cage of chabazite structure. The peak at 420 °C refers to the reduction of Fe species [47], allowing the iron reduction peaks centered at high temperature to become apparent, and the copper peaks to weaken. The broader peaks after Fe loading illustrates that Fe species had a variety of forms. This result proves that the enhancement of the redox ability of Fe species might benefit for the reduction of $NO_x$ at high temperatures.

## 2.3. Cu and Fe Species onto SAPO-34

In order to discover the identities of different Cu and Fe species, UV-Vis diffuse reflectance spectra (UV-Vis DRS) of Cu/SAPO-34, Fe/SAPO-34, and Cu-Fe/SAPO-34 were deployed, and the results are displayed in Figure 6. In the spectra of Cu/SAPO-34 sample, an observation of an intense band at around 225 nm is assigned to a d–d transition of the charge-transfer band, which is related to O → Cu transition from lattice oxygen to isolated $Cu^{2+}$ ions [21]. Besides, a broad band around 450–550 nm is attributed to the charge transfer bands of O–Cu–O and Cu–O–Cu, suggesting the presence of CuO. In the Fe-doped sample, the bands below 300 nm in the doped Fe sample represents an isolated $Fe^{3+}$, particularly tetrahedral and octahedral coordination at an absorption band of 225 nm and 265 nm, respectively. The spectrum of this sample contains several additional bands around 300–600 nm. The broad bands between 300 and 400 nm are attributed to oligomeric $Fe_xO_y$ clusters, above 400 nm is assigned to large $Fe_2O_3$ particles while that above 450 nm is considered to a d–d transition of $\alpha$-$Fe_2O_3$ [19]. For the Cu-Fe/SAPO-34 sample, the obtained result reveals that the bimetallic catalyst contained several active sites of both copper and iron. To be more specific, copper was found to be in the form of isolated $Cu^{2+}$, while iron species could be in three different types, including isolated $Fe^{3+}$, $Fe_xO_y$ oligomer, and $\alpha$-$Fe_2O_3$. It is clear to see that comparing to mono-metal catalysts, the intensity of the absorption curve in 225–550 nm increases, indicating a simultaneous improvement of the amounts of iron active sites in the sample. According to literature, the catalytic performance on the removal of $NO_x$ might be enhanced by oxo-$Fe^{3+}$ and oligomeric $Fe_xO_y$ clusters, while aggregated $\alpha$-$Fe_2O_3$ was unfavorable to $NH_3$-SCR [48,49].

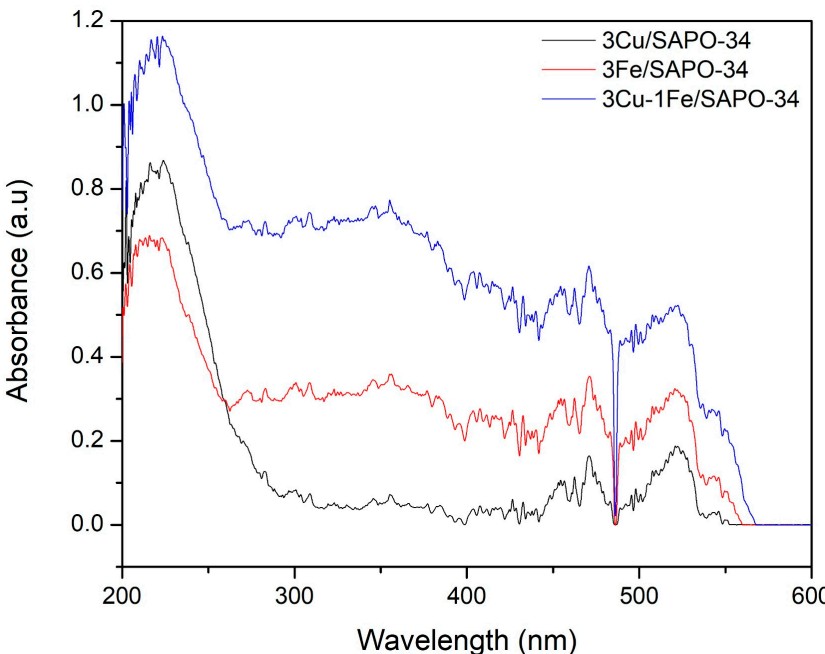

**Figure 6.** UV–Vis DRS spectra of the Cu/SAPO-34, Fe/SAPO-34 and Cu-Fe/SAPO-34 samples.

Electron paramagnetic resonance (EPR) is an effective method to identify both qualitatively and semi-quantitatively the isolated $Cu^{2+}$ or $Fe^{3+}$ ions in Cu-, Fe- based zeolite catalysts, as well as to probe their coordination environment [43,45]. The signal collected from EPR in copper samples, which normally present hyperfine structures in the parallel direction with four adsorption peaks and a sharp peak in the vertical region [50], is isolated $Cu^{2+}$. In contrast, other Cu species do not give any interactions [8,50]. This behavior is due to the copper nuclear spin (I = 3/2) and the coupling between the 3d unpaired electrons in the isolated $Cu^{2+}$ ions. As shown in Figure 7a, all copper samples show a similar axial signal of isolated $Cu^{2+}$ sites at $g_{\parallel}$ = 2.29, $A_{\parallel}$ = 131 G and $g_{\perp}$ = 2.07. These values are attributed to the octahedral coordination of $Cu^{2+}$ ions with four framework oxygen atoms in the CHA framework [8,42]. Moreover, as the axial signal of isolated $Cu^{2+}$ species is not well-resolved hyperfine structure, an isotropic signal at g = 2.17 can also be included. This isotropic signal is assigned to isolated $[Cu(H_2O)_n]^{2+}$ ions [51]. Increasing the amount of doped metal from 1 wt.% to 3 wt.% leads to a significant increase in the intensity of the axial signal, which indicates that more isolated $Cu^{2+}$ ions were formed. However, with higher Cu content of 5 wt.%, the amount of isolated $Cu^{2+}$ species decreases, and the signal becomes broader, suggesting the presence of a higher amount of isotropic signal. The broad isotropic signal might also be related to the magnetically interacting $Cu^{2+}$ species in dimeric species or such Cu sites located in close vicinity.

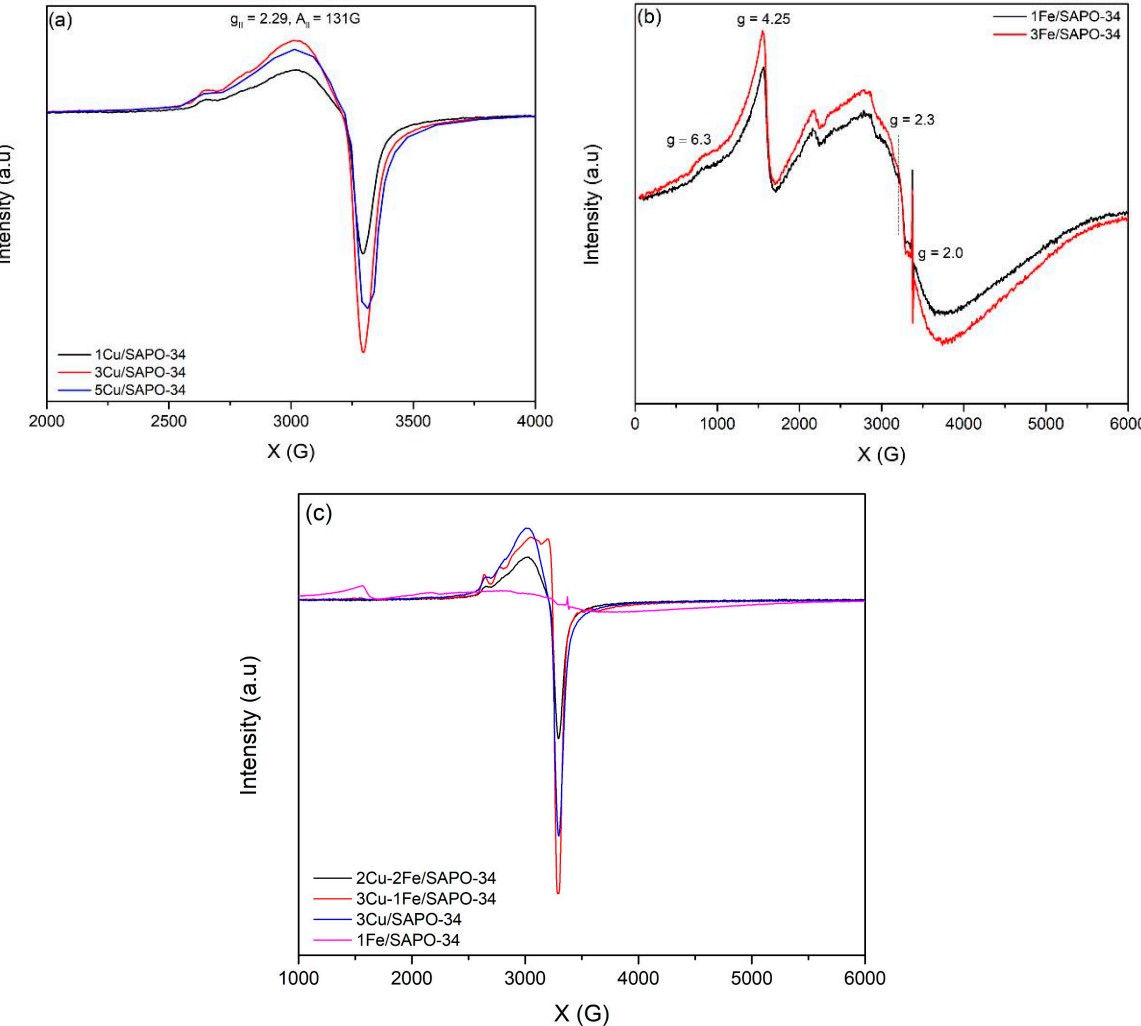

**Figure 7.** EPR spectra of (**a**) Cu/SAPO-34, (**b**) Fe/SAPO-34, and (**c**) Cu-Fe/SAPO-34 series catalysts measured at 100 K.

The EPR spectra recorded at 100 K of Fe/SAPO-34 samples are shown in Figure 7b. Three signals of different iron species are observed, including a narrow signal at g = 4.25, a low field shoulder at g = 6.3, and two broader signals at g ≈ 2.3 and g ≈ 2.0. Specifically, the signals at g = 4.25 and 6.3 both demonstrate the high spin isolated $Fe^{3+}$ species in strong rhombic and high distortion, and in higher coordination numbers, respectively. This could be caused by the isolated paramagnetic iron monomers found at ion-exchange positions inside the zeolite micropores [52]. The weak low-field line g ≈ 2.3 and 2.0 signal are probably an axial stack of distorted $Fe^{3+}$ monomers and for interacting octahedrally with residual $Fe_2O_3$ phase [52]. It is clear to see that with an increase in the amount of Fe loading, especially in samples possessing iron higher than 1 wt.%, signals at g = 4.25, 2.3, and 2.08 become more intense. Moreover, the broader signal at g ≈ 2.3 and g ≈ 2.0 in the samples with higher Fe content (3 and 5 wt.%) indicates an increase in the amount of clustered Fe oxides in these samples.

As shown in Figure 7c, comparing to the mono-metal catalysts, EPR signals of bimetallic catalysts exhibit dominant axial signals from isolated $Cu^{2+}$ sites. Compared to the sample 3Cu/SAPO-34, the intensity of the axial $Cu^{2+}$ signal in sample 3Cu-1Fe/SAPO-34 is higher, indicating that the introduction of an appropriate amount of Fe led to increasing the number of isolated $Cu^{2+}$ ions. As lower Cu content and higher Fe content, the intensity of the axial $Cu^{2+}$ signal from 2Cu-2Fe/SAPO-34 sample is lower than that of 3Cu-1Fe/SAPO-34. In other words, this demonstrates that fewer active isolated $Cu^{2+}$ ions were formed after decreasing the amount of Cu loading. No signal from isolated $Fe^{3+}$ species (g ≈ 4.3 and g ≈ 6.3) can be seen in the EPR spectra of bimetallic catalysts. However, the presence of $Fe_xO_y$ clusters cannot be excluded since their signals at g ≈ 2.3 and 2.0 might be superimposed on the axial $Cu^{2+}$ signal.

### 2.4. Catalyst Performance

Figure 8 shows the effect of temperature on $NO_x$ conversion over all catalysts. For the Cu/SAPO-34 samples (Figure 8a), the $NO_x$ conversion improves significantly from 150 to 250 °C and remains above 80% between 250 and 550 °C, especially in the region of 300–450 °C when the yield reaches up to more than 90%. Afterward, above 400 °C, the $NO_x$ conversion slowly declines and reaches only 78% at 600 °C. Among copper-loaded samples, the sample with 3 wt.% Cu shows the highest activity.

It is clear from the catalytic performance of Fe/SAPO-34 samples (Figure 8b) that the incorporation of iron onto SAPO-34 slightly enhances the SCR reaction in the whole temperature range. However, a good performance could only be achieved in a narrow temperature range of 350–600 °C. Higher Fe content (up to 5 wt.%) $NO_x$ conversion tends to slightly decrease at high temperatures. These results demonstrate that more metals loading could have disadvantage effects on $NO_x$ conversions. Therefore, the selection for the content of the doped metal is crucial for enhancing the $NO_x$ removal yield. In summary, the results of Figure 8a,b suggest that suitable metal content could promote $NO_x$ conversion, while the excessive Cu or Fe loading could block the "channel" of zeolites. This could be examined by the result of $N_2$ adsorption-desorption in Figure 2 and Table 1 above.

When incorporation of both Cu and Fe into SAPO-34, both 3Cu-1Fe/SAPO-34 and 2Cu-2Fe/SAPO-34 samples significantly enhance the $NO_x$ conversion in the broader temperature range of 200–600 °C. In comparison with these two samples, the results in Figure 8c show that the sample 3Cu-1Fe/SAPO-34 displays the best catalyst. Comparing this bimetallic sample to other mono-metal catalysts that possess good catalytic activity, a similar result was obtained as it could be seen in Figure 8c. In particular, compared to 3Cu/SAPO-34, $NO_x$ conversion over the 3Cu-1Fe/SAPO-34 catalyst decreases slightly at low temperatures (below 200 °C), then an increase of 7% is observed in the range of 200–450 °C, while that in between 550 and 650 °C is 8%. Notably, the $NO_x$ removal yields still reached 95% at 600 °C, indicating that Cu-Fe/SAPO-34 catalysts have enlarged the operating temperature window.

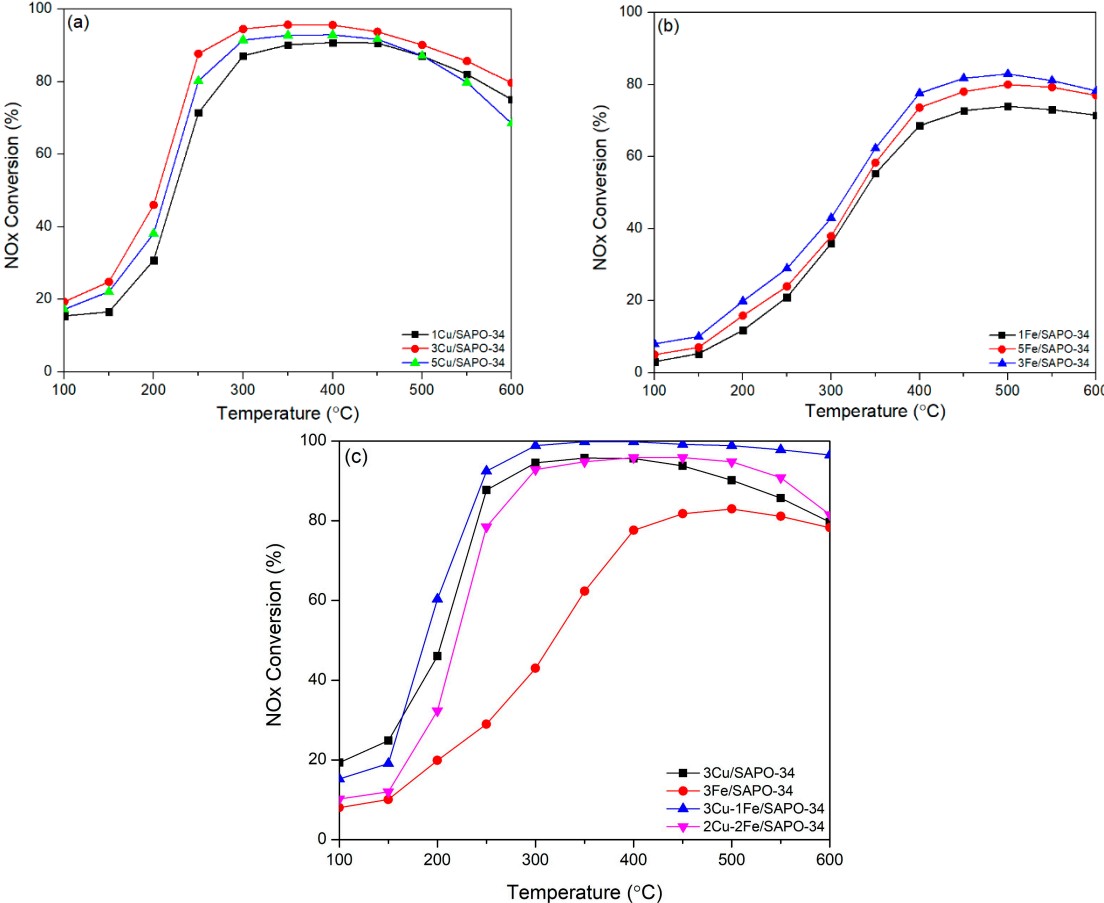

**Figure 8.** $NO_x$ conversion during standard $NH_3$-SCR as a function temperature of the (**a**) Cu/SAPO-34, (**b**) Fe/SAPO-34, (**c**) Cu-Fe/SAPO-34 and comparison of all catalyst samples.

It is known that the decrease of $NO_x$ conversion at high temperatures together with the decline of $N_2$ selectivity, which is due to the oxidation of $NH_3$ to undesired $N_2O$ formation. Therefore, the $NH_3$ oxidation activity of the selected catalysts was also conducted. As shown in Figure 9, three samples share the same curve shapes of either $NH_3$ conversion, or $NO_x$ concentration. The ammonia oxidation reaction initiates at 150 °C but occurs with a low rate in the temperature range below 350 °C. Above this temperature, the $NH_3$ conversion increases sharply. Sample with singular copper shows the highest $NH_3$ oxidation activity, followed by 3Fe/SAPO-34 sample, while the bimetallic sample exhibits the lowest in this activity. To be more specific, the conversion of Cu/SAPO-34 was nearly 100% at 600 °C, while that in the iron-loaded sample was around 95%. The total amount of $NO_x$ formed during $NH_3$ oxidation in all four samples was less than 8 ppm for the whole investigated temperature range, indicating a high selectivity of $N_2$ in the $NH_3$ oxidation reaction.

The reaction steps occurring on the catalyst mainly include the standard SCR and $NH_3$ oxidation reactions as follows:

$$4\,NO + 4\,NH_3 + O_2 \rightarrow 4\,N_2 + 6\,H_2O \tag{1}$$

$$4NH_3 + 3O_2 \rightarrow 2N_2 + 6H_2O \tag{2}$$

As shown in Figures 8 and 9, the performance of samples on standard $NH_3$-SCR reaction was improved considerably than that on the $NH_3$ oxidation reaction below 350 °C, suggesting that the $NH_3$ oxidation reaction contributes a slight effect on the catalytic activity in this temperature range. For the sample 3Cu-1Fe/SAPO-34, a strong ammonia inhibition effect of iron leads to limited $NH_3$-SCR performance below 350 °C [53]. However, the catalytic activity of this catalyst displayed an

improvement in the range of 200–450 °C, which could be explained by the increase in the amount of isolated $Cu^{2+}$ ions (Figures 6 and 7) that are reported to be active sites for SCR reaction [50].

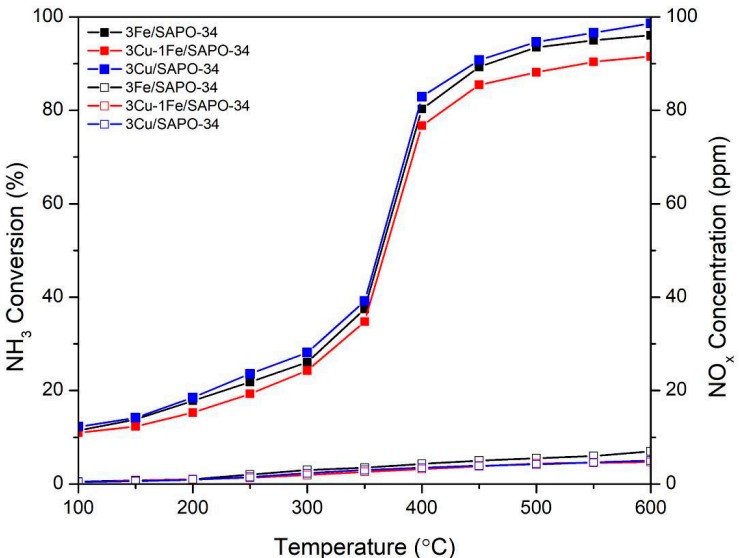

**Figure 9.** $NH_3$ conversion (filled symbols) and $NO_x$ concentration (open symbols) as a function of temperature during the $NH_3$ oxidation experiment for the Cu/SAPO-34, Fe/SAPO-34, and Cu-Fe/SAPO-34 samples.

From Figure 9, in the temperature range above 350 °C, precisely above 450 °C, the $NH_3$ oxidation occurs significantly. In the case of the Fe-doped catalyst, $Fe_xO_y$ oligomers and $Fe_2O_3$ particles were attributed as the promoters for the $NH_3$ oxidation [54]. Similarly, in our case, an increase in the amount of these iron species in the bimetallic sample is suitable for a higher yield of $NH_3$ oxidation. However, the presence of the $Fe_2O_3$ clusters and $Fe_xO_y$ species in the Cu-Fe/SAPO-34 sample could not enhance its $NH_3$ oxidation activity when compared to the Cu/SAPO-34 sample. This could be explained by the decrease in the number of bulk CuO species in the bimetallic sample, which are assumed to be highly active for $NH_3$ oxidation [28]. Therefore, a reduction in the amount of bulk CuO species on the surface, by incorporating Fe with Cu inhibits the $NH_3$ oxidation reaction. Hence, the decrease in $NH_3$ oxidation ability and the enhancement of the small amount of oligomeric $Fe_xO_y$ species and isolated $Cu^{2+}$ contribute to the improvement in the catalytic activity of the 3Cu-1Fe/SAPO-34 sample.

For the sample 2Fe-2Cu/SAPO-34, as discussed at EPR results, a small amount of isolated $Cu^{2+}$ ions are altered by the oligomeric $Fe^{3+}$ ions while the isolated $Cu^{2+}$ ions combined easily to the $CuO_x$ species [21,55]. Thus, the reduction in the number of isolated $Cu^{2+}$ (Figure 7c) decreases the SCR activity of the sample 2Fe-2Cu/SAPO-34 in 200–450 °C and the increased amount of oligomeric $Fe_xO_y$, $Fe_2O_3$ clusters in the sample resulted in a decline at the high-temperature activity at above 450 °C. Meanwhile, the decrease in SCR performance below 200 °C could be a consequence of the reduction in the surface area and pore volume of SAPO-34, which are caused by the aggregated $Fe_2O_3$ particles [42].

The hydrothermal stability of catalysts is an important issue in SCR catalysts since the heat generating during DPF regeneration process need to be dealt with [17]. In order to investigate this property, the 3Cu/SAPO-34 and 3Cu-1Fe/SAPO-34 catalysts were treated in 10% $H_2O$/air under a total flow rate of 1 l/min at 750 °C and 850 °C for 36 h. The results on the SCR activity between the fresh and hydrothermal aging samples are exhibited in Figure 10a. $NO_x$ conversion of hydrothermally treated samples decreases during the whole temperature range, especially at high temperatures. Furthermore, this deNO$_x$ activity of the 3Cu-1Fe/SAPO-34 sample decreased when the aging temperature was increased. To be more specific, both 3Cu-1Fe/SAPO-34_750 and 3Cu-1Fe/SAPO-34_850 achieved more than 85% $NO_x$ conversion from 250–500 °C. Compared to 3Cu/SAPO-34_750, both of Cu-Fe/SAPO-34 samples with two hydrothermal aging conditions show significantly higher $NO_x$ conversion, indicating

that Cu-Fe/SAPO-34 is more robust and resistant to relatively harsh hydrothermal treatment than Cu/SAPO-34. To be more specific, XRD patterns of the fresh and hydrothermal aging catalysts were performed through Figure 10b. The result deconvoluted that all the other catalysts remained the CHA zeolite structure apart from the 3Cu/SAPO-34 sample aged at 850 °C for 36 h. This singular copper exchanged SAPO-34 catalyst appeared to be more sensitive to hydrothermal treatment since the collapse of the CHA structure occurred at 850 °C. This framework damage is probably due to the aggregation of copper species, which could negatively affect the stability of the structure and lead to the deterioration in the catalytic activity on $NH_3$-SCR [54]. On the contrary, the hydrothermal treatment at 750 °C for 36 h or even at harsher conditions just had a negligible effect on the structure of 3Cu-1Fe/SAPO-34 samples. That insignificant influence on the bimetallic catalyst indicates that the introduction of iron could help to stabilize the copper dispersion during aging, which can also be observed in the XRD patterns of fresh samples (Figure 1). Therefore, the improvement in dispersing copper appears to be the vital reason for the enhancement of the hydrothermal stability of the Cu-Fe/SAPO-34 catalyst [16].

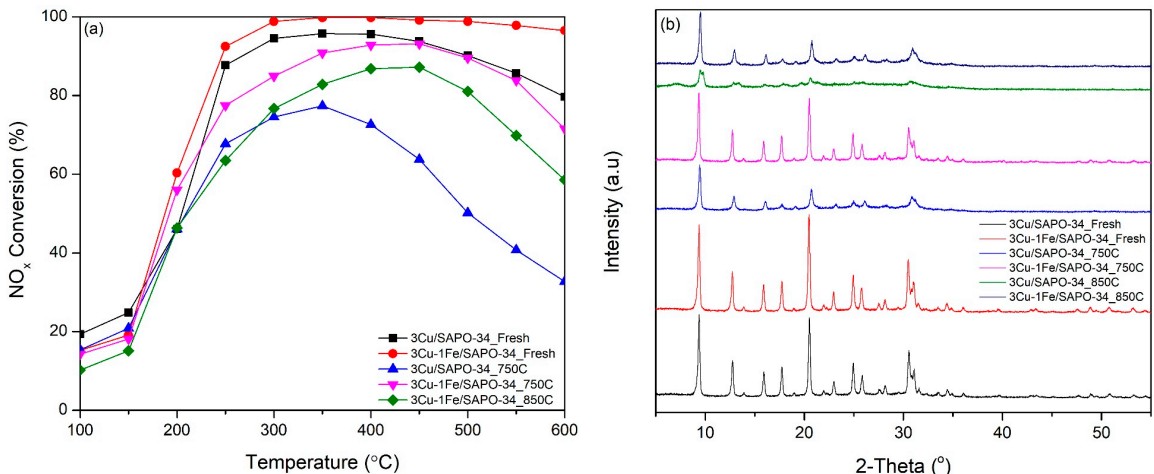

**Figure 10.** (**a**) $NO_x$ conversion of Cu/SAPO-34 and Cu-Fe/SAPO-34 after hydrothermal aging and (**b**) XRD patterns of fresh and hydrothermal aging catalysts.

## 3. Materials and Methods

### 3.1. Catalyst Preparation

According to our previous study, SAPO-34 molecular sieves were synthesized from a reaction mixture with a molar composition of 1 $Al_2O_3$:0.6 $SiO_2$:1 $P_2O_5$:3 TEA:3 Mor:1 TEAOH:110 $H_2O$ [29]. Aluminum isopropoxide (98%, Merck), tetraethyl orthosilicate (TEOS, 98%, Sigma), and phosphoric acid (85%, aqueous solution, Merck) were used as aluminum, silicon, and phosphorus precursors, respectively. Morpholine (Mor, 99%w/w, ACS Reagent, Sigma), tetraethylammonium hydroxide (TEAOH, 25%w/w, Sigma), triethylamine (TEA, 98%, Merck) were the organic structure-directing agents. Initially, aluminum isopropoxide was stirred in the distilled water and followed by adding slowly phosphoric acid. The solution was constantly agitated for 1 h to obtain a uniform mixture. Afterward, TEOS was added dropwise, and the slurry was stirred continuously for another 1 h. In the next step, the organic structure-directing agents were supplied slowly to the solution. The resulting gel was agitated for 6 h and then aged at ambient temperature for another 12 h in a stainless-steel autoclave. For the crystallization process, the mixture was then heated for 48 h at 200 °C. After that, the solid part was separated and washed with distilled water by centrifuge for several times before putting into a drying oven at 120 °C for 6 h to get rid of water. The final product was obtained by calcining at 550 °C for another 6 h.

Metals (Cu and/or Fe) were introduced to the SAPO-34 by following the two-step liquid ion-exchanged procedure. In the first step, SAPO-34 was changed into ammonium form by exchanging

with ammonium acetate solution twice. The first exchange time was carried out by constantly agitating SAPO-34 zeolites with ammonium acetate solid and distilled water at 60 °C for 3 h. The solid part was then collected by centrifugation. After that, the same procedure was repeated one more time but with a difference in exchanging time (18 h) and stirring temperature (room temperature). The centrifuged solid part was finally dried at 90 °C to achieve $NH_4^+$/SAPO-34 powders. In the second step, the ammonium form of SAPO-34 was ion-exchanged with metals. Cu and Fe were doped using $Cu(CH_3COO)_2 \cdot H_2O$ (Sigma-Aldrich) and $Fe(CH_3COO)_2 \cdot 4H_2O$ (Sigma-Aldrich), respectively. The mixture was stirred continuously at 60 °C in 24 h and followed by centrifuged and washed with distilled water to get a solid part. This collected part was then dried at 90 °C for 18 h and calcined in the gas of 20% $O_2$/Ar at 550 °C for 5 h. The SAPO-34 catalysts samples were denoted as xCu/SAPO-34, yFe/SAPO-34, and xCu-yFe/SAPO-34 in which x and y are the weight percentage of Cu and Fe used in synthesis, respectively.

### 3.2. Characterization

The X-ray diffractograms of samples were obtained by a powder X-ray diffractometer Bruker D8 (Billerica, MA, USA) at room temperature using Cu anode and Kα radiation at λ = 1.54 Å in the ranging scan from 5 to 55°, with a stepwise of 0.02°. From the diffractograms of the as-prepared materials, their average crystallite sizes were determined from Scherrer equation d = 0.9λ/βCosθ, in which d is the crystallite, λ is the wavelength of X-ray radiation and β is the full width at half maximum in radians. The morphology of catalysts was detected by the field emission scanning electron microscope (FE-SEM) on a Hitachi S-4800 (Tokyo, Japan). The chemical composition of the catalysts was determined by the Hitachi S-4800 model scanning electron microscope integrated with a dispersive energy X-ray spectrometer. The nitrogen adsorption-desorption measurements were conducted on a Micromeritics ASAP 2020 analyzers (Norcross, GA, USA) at 77 K. Before this analysis, the samples were degassed under vacuum at 300 °C. The total surface area was analyzed based on the Brunauer–Emmett–Teller (BET) theory, while the external surface area, micropore surface area, and volume were determined by using the t-plot method. UV-Vis diffuse reflectance spectroscopy (UV-Vis DRS) was conducted on Avantes Avaspec-ULS2048XL-EVO spectrometer (NS Apeldoorn, Netherlands) equipped with an integrating sphere coated with $BaSO_4$. AvaSoft 8 software (NS Apeldoorn, Netherlands) was used for deconvolution of the UV-Vis spectra collected from 180 to 1100 nm. The various iron species were quantified relative to each other by the area ratios of the corresponding sub-bands [51].

Temperature-programmed desorption with ammonia ($NH_3$-TPD) experiments were performed by a Micromeritics Auto Chem 2920 instrument (Norcross, GA, USA). In all measurements, 0.1 g of the samples were put in a quartz fixed-bed U-shaped microreactor and outgassed in a constant flow of 50 mL/min helium gas for 1 h at 300 °C. Thereafter, the samples were cooled down to 100 °C and followed by purging to $NH_3$ flow (10% $NH_3$/He) with the flow rate of 30 mL/min for 1 h. After that, they were exposed with a helium flow (50 mL/min) at 100 °C for 1 h to get rid of physisorbed ammonia. The procedure was repeated until a stable baseline was acquired. Finally, the $NH_3$ chemisorbed samples were heated from 100 to 600 °C at a fixed heating rate of 10 °C/min in a helium flow (50 mL/min). The signal from the ammonia desorption was then recorded. The temperature-programmed reduction with $H_2$ ($H_2$-TPR) was conducted with the same equipment of $NH_3$-TPD measurements and the following procedure. For the pretreatment of the catalysts before the measurements, helium gas was blown into the quartz fixed-bed U-shaped microreactor, in which 0.1 g of the catalyst was held firmly in place, at 500 °C for 1 h and then cooled to 50 °C. The heating program of the TPR experiments included a ramp of 10 °C/min in 10 vol% $H_2$/He from 50 to 900 °C with a total gas flow 30 mL/min. The temperature and detector signals were then continuously recorded while heating at 15 C/min up to 900 °C. The $H_2O$ and $CO_2$ produced during the experiments were separated by a cooling trap placed between the sample and the detector.

Electron paramagnetic resonance (EPR) spectra were recorded with a continuous wave X-band by Bruker EMX-Micro EPR spectrometer (Billerica, MA, USA) with a microwave power of 0.188 mW,

modulation frequency 100 kHz and amplitude 2 G at 100 K, the magnetic field was full range from 100 to 6600 G. The g-value was determined from precise frequency and magnetic field values, which was shown automatically on EPR console.

### 3.3. Catalytic Activity

The catalytic performance in the NH$_3$-SCR reaction was carried out by ABB Gas Analyzer AO2020 Limas21 (UV sensor) spectrometer (Zürich, Switzerland) in a fixed-bed quartz reactor running in a steady flow mode (Figure 11). To avoid condensation along the upstream tube, all the gas lines were heated and maintained about 120 °C. In each experiment, 100 mg catalyst (40–60 mesh) and 200 mL/min in total flow rate was utilized, corresponding to a gaseous hourly space velocity (GHSV) of 120 000 h$^{-1}$. The reactant gas included 1000 ppm NO, 1000 ppm NH$_3$, 8 vol.% O$_2$ and Ar as balance gas. Prior to the measurement, the catalyst was activated in 20% O$_2$/Ar flow for 2 h at 550 °C. The reaction temperature was raised stepwise from 100 to 600 °C. The typical time to achieve steady-state at each temperature was about 1 h. For the NH$_3$ oxidation reaction, a similar condition was followed where NO was added in the gas mixture. In the investigation of hydrothermal stability, the catalysts were aged in a quartz tube reactor at 750 °C or 850 °C in 10% H$_2$O/air under a total flow rate of 1 l/min for 36 h before the activity test. The conversions of NO$_x$ and NH$_3$ are determined using the following formula dependent on the concentrations of inlet and outlet gas at a steady-state:

$$NO_xConversion = \frac{C_{NO_x inlet} - C_{NO_x outlet}}{C_{NO_x inlet}} \times 100\%$$

$$NH_3Conversion = \frac{C_{NH_3 inlet} - C_{NH_3 outlet}}{C_{NH_3 inlet}} \times 100\%$$

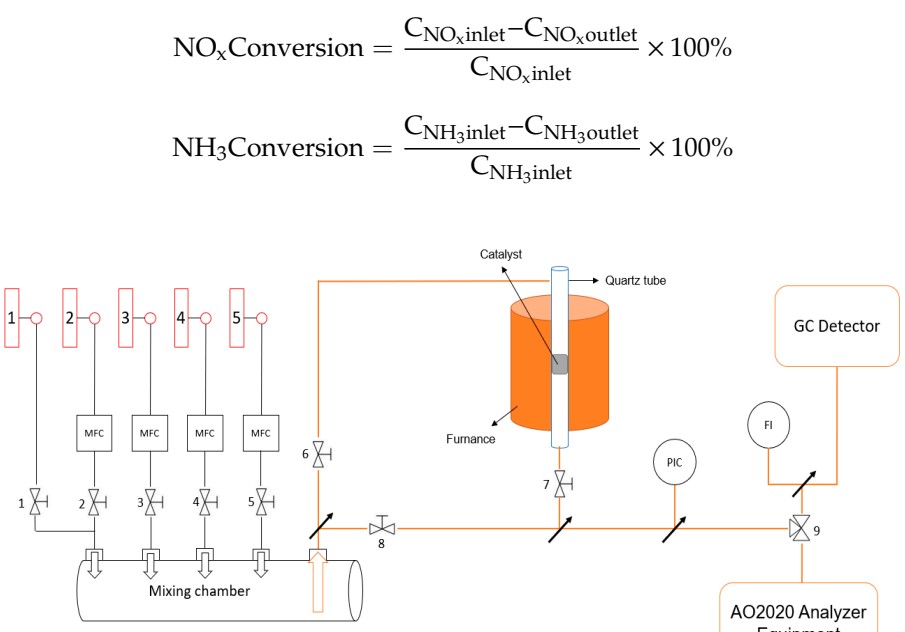

**Figure 11.** Schematic diagram of the experimental apparatus for activity test.

## 4. Conclusions

The Cu/SAPO-34, Fe/SAPO-34, and bimetal Cu-Fe/SAPO-34 catalysts were synthesized via a liquid ion-exchange method, and their application for NH$_3$-SCR reactions was investigated. As seen from the XRD patterns, the addition of Cu or Fe does not affect the chabazite structure of SAPO-34 crystal. The synthesized Cu-, Fe- based SAPO-34 catalysts crystallized in cubic structures with the same average sizes; however, the morphology of the SAPO-34 crystals could be changed with the existence of high Cu or Fe content. The result of N$_2$ adsorption-desorption measurement indicated a steady decrease in the surface area, resulting from the higher content of metals. The micropores also declined after ion-exchanging. XRD, N$_2$ adsorption-desorption, and FE-SEM also proved that most of the crystal and physical structure of SAPO-34 remained when co-doping Cu and Fe together. H$_2$-TPR results demonstrated that Cu-Fe/SAPO-34 also had good redox ability in a broader temperature range

compared to Cu/SAPO-34 because of the high-temperature redox capacity and rich forms of iron components. Using the UV-Vis DRS, different active sites, namely, $Fe^{3+}$ and $Fe_xO_y$ after Fe loading, and a strong interaction between Fe and Cu were observed. This is also supported by EPR results. Comparing to the Cu/SAPO-34 catalyst, which contains the same amount of copper loading, the quantity of isolated $Cu^{2+}$ was improved when using an appropriate amount of Fe in the Cu-Fe/SAPO-34 sample. Moreover, an additional amount of oligomeric $Fe_xO_y$ in this sample benefits of the $NH_3$-SCR reaction at high temperatures. $NH_3$-TPD results revealed that the addition of Fe or Cu enhanced the acid sites, therefore, brought about benefits to the $NH_3$ adsorbed, as well as widening the catalytic reaction temperature range.

Co-doping of Cu-Fe into SAPO-34 led to higher catalytic performance for $NH_3$-SCR, caused by the synergistic impacts between iron and copper. To be more specific, the bimetallic zeolite's advantage went beyond the parent mono-metallic zeolite when it came to the molar ratio of Cu to Fe, reaching nearly value 3. Since the $Fe_2O_3$ particles would block the pores of the Cu-Fe/SAPO-34, the $NO_x$ conversion in low-temperature range below 150 °C slightly dismissed. Thanks to the raised amount of isolated active $Cu^{2+}$ sites, the activity for the bimetal Cu-Fe sample in 200–350 °C upturned. Besides, the less amount of surface bulk CuO species got, the more decrease the $NH_3$ oxidation above 350 °C showed. The enhanced SCR performance of Cu-Fe/SAPO-34 samples above 350 °C resulted from both the reduced $NH_3$ oxidation activity and the additional oligomeric $Fe^{3+}$ active sites. Although hydrothermal stability enhanced over Cu-Fe/SAPO-34 compared with Cu/SAPO-34 at 750 °C and 850 °C, the improvement of catalysts with poisoning $SO_2$ and hydrocarbon tolerance necessitated the need for further research.

**Author Contributions:** Data curation, T.D.; Formal analysis, T.D.; Funding acquisition, T.H.V. and T.H.P.; Methodology, T.D.; Project administration, T.H.P.; Supervision, M.T.L., T.H.V. and T.H.P.; Writing—original draft, T.D., P.D. and K.N.; Writing—review & editing, M.T.L. and T.H.V. All authors have read and agreed to the published version of the manuscript.

**Funding:** This research was funded by the Vietnam National Foundation for Science and Technology Development (NAFOSTED) under the grant number 104.05-2018.306. This work also assisted by RoHan Project funded by the German Academic Exchange Service (DAAD, No. 57315854) and the Federal Ministry for Economic Cooperation and Development (BMZ) inside the framework "SDG Bilateral Graduate school programmed.

**Acknowledgments:** This work has been supported by the Vingroup Innovation Foundation (VINIF) under the grant number VINIF.2019.TS.74.

**Conflicts of Interest:** The authors declare no conflict of interest.

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
