# Peer review of "Copper-Iron Bimetal Ion-Exchanged SAPO-34 for NH3-SCR of NOx"

_catalysts, doi:10.3390/catal10030321_

Round 1

Reviewer 1 Report

Manuscript ID: catalysts-734585

It was my pleasure to review work entitled: “Copper-Iron Bimetal Ion-Exchanged SAPO-34 for NH3-SCR of NOx”. Article describes in the detail the synthesis, characterisation and evaluation of catalytic activity of Copper-Iron Bimetal Ion-Exchanged SAPO-34 used for the selective catalytic reduction of NOx with NH3.

Article is written in suitable English making this work easy to follow and understandable. Authors performed very professional comparison of their Cu-Fe SAPO-34 catalyst with already known and described in the literature Cu and Fe SAPO-34 catalysts.

Authors presented in the main text interesting data about how the synthesis conditions influence the structural (XRD, gas adsorption) and textural (SEM) properties of the obtained of materials. Article presents full physicochemical characteristic of the synthesised catalysts. Author’s conclusions are supported by the scientific data and experiments (NH3-TPD, H2-TPD, UV–Vis DRS, EPR) clearly described in the experimental part of the article.

Authors have established the catalyst performance at wide temperature range and confirmed that Cu-Fe SAPO 34 allows to broaden the operating temperature window maintaining high conversion of 95%.

Part of work which is missing is the effect of poisoning of catalyst with SO2 or hydrocarbons tolerance and the catalysts tolerance/resilience to the mentioned substances.

I have only one remark/question regarding this work:

  • Have you done any elemental analysis or structural analysis like XPS, EDXS or XRD of your electrodes after the 36h of hydrothermal treatment? I think it would be interesting to observe the changes which are happening after the aging.

For the mentioned reason I recommend this work to undergo minor revision.

Author Response

We thank the reviewer #1 for appreciating our manuscript. Please see the attachment file below.

Reviewer 2 Report

This manuscript contains novel findings on the catalytic effect of Cu-Fe/SAPO-34 for NOx reduction. The subject of this study is appropriate for publication in Catalysts. Overall, the results are quite informative. Several minor revisions described below are recommended prior to acceptance for publication. 

1) Figure 1, line 120–121: The XRD peak intensity of Cu-Fe/SAPO-34 was reported to be greater than that of Cu/SAPO-34, but the intensities cannot be compared directly because the XRD patterns of Cu-Fe and Cu samples are separated (Fig. 1(a) and (c)). The XRD patterns to be compared are better to be shown in same figure, otherwise the numerical ratio of their intensities should be provided.

2) line 130–136: The authors described that, if the metal loading is relatively small, the BET specific surface area and averaged pore diameter of metal-loaded catalysts do not markedly change from those of pristine SAPO-34 while they are significantly influenced if the metal loading reaches 5%. However, the values listed in Table 1 look as if they are inconsistent with the description. For example, the single-metal samples were prepared to make 1, 3, and 5% metal loading, whereas the metal loading of the mixed-metal samples were 2+2% and 3+1% (total 4%); the single- and mixed metal samples cannot be simply compared. Additional comments on the data interpretation should be added to help understanding. In addition, since these data affects the discusstion of the later section, the discussion of the later part should be reconfirmed.

3) Figure 3, line 153–158: The morphological features of the samples were reported based on their coverage with random-shaped materials and surface roughness. However, the morphological difference of the samples cannot be clearly recognized from Figure 3. Also, the difference may be accidentally observed one. Additional explanation on the difference should be added to justify the conclusion.

4) Figure 9: The ammonia conversion and NOX concentration are included in one fugure. As I see, the open and filled markers indicate the NOx concentration and NH3 conversion, respectively; they should be noticed in the figure caption. 

Author Response

We thank the Reviewer #2 for all comments. We have answered point-by-point in Response File below. Please see the attachment

Reviewer 3 Report

See comments in attached file.

Author Response

We thank the Reviewer #3 for all comments and suggestion. We have answered point-by-point in Response File below. Please see the attachment

Reviewer 4 Report

This is an interesting manuscript describing an experimental study of mono and bi-metallic catalysts supported on SAPO-34 used for selective catalytic reduction.  The work could be a useful contribution to the literature, but a number of improvements in clarity and English presentation are required prior to publication.  Therefore, I recommend minor revision.

The Abstract should be strengthened and made more quantitative.  Currently, it contains a long list of characterizations and vague adjectives, such as “high” to describe the conversion over a “broad temperature window”.  I would prefer to see a more concrete description of the key results, such as “for the catalyst with 3 wt% Cu and 1 wt% iron, ~100% conversion of 1000 ppm NO was achieved at reaction temperatures from 350 to 500°C at a space velocity of 120,000 h-1.”  This sentence is just a suggestion that the authors can surely improve on, but the main point is to make the Abstract more descriptive, quantitative, and useful to the reader.

In line 142, the authors state that there is “a small hysteresis” present in the data of Figure 2.  This is not at all clear from the figure.  If the authors wish to make this point, they need to expand the y-axis of the figure to show that there actually is hysteresis in the curves in the figure.

The discussion regarding Figure 3 in the paragraph in lines 146-161 need to be clarified.  When the authors reference “random-shaped materials”, it looks like all of the images contain this.  Please be more descriptive and accurate to explain the differences among the figures.

The statement in lines 221-222 that “This result proves that the enhancement of the redox ability of Fe species could improve the high-temperature conversion of NO . . .” is too strong.  I do not believe that the presented TPR data have “proven” this.  Please reword or provide stronger evidence of the proof.

Lines 273-274 contain the phrase “bimetallic catalysts exhibit almost Cu2+ signal”.  What does this mean?  Almost the same as the signal from Cu2+ or something else?  Please clarify.

In lines 291-292, iron is said to enhance the SCR reaction over the whole temperature range in Figure 8b, but this is not clear, especially compared to the other catalysts that have much larger effects at lower temperatures.  Please explain this more clearly.

Figure 9 contains two y-axes, but the data belonging to the axes are not clearly identified.  Please clearly identify which data belong to which axis (presumably from the text, the open squares are associated with the right-hand axis).

In line 440, the line temperatures are said to be “almost maintained at 120°C”.  What does this mean?  Either it was 120°C or something other lower temperature, but if it was a lower temperature, a more accurate description should be used (e.g., “at least 115°C” or “about 120°C”).  Please make this more accurate.

The statement in line 469 that the “catalyst sample while still kept the total Cu amounts remained” is not understandable.  Please revise.

The English quality of the manuscript should be improved.  As a few of the many items that I noticed:

In lines 20 and 21, the phrase “samples were further characterized physicochemical properties” is apparently missing a word or more.

In line 68, what is meant by “small content of Fe/ZSM-5 piece”?  Please clarify or describe more accurately.

In line 105, I am not sure what “sides” means in this context.  Please use a different, more accurate word.

In line 109, the word “performed” is used when “tested” would seem to be correct.

In line 125, the phrase “below the concentrations of them” is used referring to the lack of XRD peaks for the high metals loading.  Are the authors suggesting that the metals were not successfully loaded at this concentration?  The EDX data indicate that they were loaded at this level, so this statement should not be made or the authors should clarify their intended meaning.

Lines 130-131 contain the sentence, “It can be observed that both BET surface area and pore size almost remain when the metal loading varies . . . .”  This appears to be missing a word:  “almost remains constant”?  Please correct (although I would not agree that these almost remain constant, so the authors need to clearly state what they mean here).

The start of line 138 appears to have an extraneous “it” (“Furthermore, XRD results, it indicates . . .”).

The phrase “The next two centered can be attributed . . .” in line 171 appears to be missing a word.

The sentence in lines 186-189 (“In our situation, the combination of the Fe(CH3COO)2.4H2O with the Brønsted acid protons in the Cu/SAPO-34 make the catalyst in the acidic aqueous environment, resulting the formation of extra-framework Cu2+ from a part of bulk CuO [37]”) needs to be clarified.  Are the authors saying that the synthesis environment is acidic?  If so, “make the catalyst in the acidic aqueous environment” does not state this clearly.

In line 231, the word “obtains” is used when “contains” appears to be correct.

Author Response

We thank the Reviewer #4 for all comments. We have answered point-by-point in Response File below. Please see the attachment

Round 2

Reviewer 3 Report

Manuscript can be published in present form.